# Advances in Mixer Design and Detection Methods for Kinetics Studies of Macromolecular Folding and Binding on the Microsecond Time Scale

**DOI:** 10.3390/molecules27113392

**Published:** 2022-05-25

**Authors:** Takuya Mizukami, Heinrich Roder

**Affiliations:** Molecular Therapeutics Program, Fox Chase Cancer Center, Philadelphia, PA 19111, USA; Takuya.Mizukami@fccc.edu

**Keywords:** turbulent mixing, continuous flow, fluorescence, reaction mechanism, protein folding, protein–ligand interactions

## Abstract

Many important biological processes such as protein folding and ligand binding are too fast to be fully resolved using conventional stopped-flow techniques. Although advances in mixer design and detection methods have provided access to the microsecond time regime, there is room for improvement in terms of temporal resolution and sensitivity. To address this need, we developed a continuous-flow mixing instrument with a dead time of 12 to 27 µs (depending on solution viscosity) and enhanced sensitivity, sufficient for monitoring tryptophan or tyrosine fluorescence changes at fluorophore concentrations as low as 1 µM. Relying on commercially available laser microfabrication services, we obtained an integrated mixer/flow-cell assembly on a quartz chip, based on a cross-channel configuration with channel dimensions and geometry designed to minimize backpressure. By gradually increasing the width of the observation channel downstream from the mixing region, we are able to monitor a reaction progress time window ranging from ~10 µs out to ~3 ms. By combining a solid-state UV laser with a Galvano-mirror scanning strategy, we achieved highly efficient and uniform fluorescence excitation along the flow channel. Examples of applications, including refolding of acid-denatured cytochrome c triggered by a pH jump and binding of a peptide ligand to a PDZ domain, demonstrate the capability of the technique to resolve fluorescence changes down to the 10 µs time regime on modest amounts of reagents.

## 1. Introduction

Fast time-resolved measurements are essential for gaining mechanistic insight into biological processes such as enzyme catalysis, protein and RNA folding and protein–ligand interactions, which often occur on a time scale extending into the microsecond range. Various techniques have been used to trigger reactions, including temperature and pressure jumps [1,2,3], flash photolysis [4], electron transfer [5], hydrodynamic focusing under laminar flow conditions [6,7] and turbulent mixing [8,9,10,11]. Flash photolysis and temperature jump techniques have been used to trigger protein folding reactions within a few microseconds or less, but they have been limited in scope because of the requirement of a photochemical trigger and cold-denatured initial state, respectively. Elegant hydrodynamic focusing experiments have been developed that can achieve about 90% efficiency for mixing a macromolecule and a small molecule (e.g., denaturant or acid) in less than 10 µs [7,12], but they have not been applied widely due to the high protein concentrations and challenging laser optics needed for monitoring reaction progress in a thin (<1 µm) layer of protein solution. In contrast, rapid mixing of two (or more) reactants by turbulent mixing has proven to be a more generally applicable and versatile approach for initiating biomolecular reactions [11,13,14,15].

The time resolution of a rapid mixing instrument is limited by the instrumental dead time, i.e., the time delay between the onset of mixing and the first reliable instrumental reading of reaction progress, which in practice is determined empirically by extrapolation of a first-order reaction back in time [16,17]. Factors contributing to the dead time include mixing efficiency, which determines the time required to achieve (nearly) complete mixing of the reactants, flow rate and dead volume between the mixing region and first point of observation. The venerable stopped-flow technique [18,19,20], coupled with a wide range of detection methods, including absorbance, fluorescence emission, fluorescence lifetime, circular dichroism, light scattering and small-angle X-ray scattering, has long been the main source of kinetic information and played a central role in elucidating kinetic mechanisms of chemical reactions, protein folding, protein–protein interaction and enzyme catalysis. Major strengths of the stopped-flow technique include its versatility, relative sample economy and wide time window, typically ranging from a few milliseconds to minutes (limited by diffusion across the mixer and convection artifacts). However, the time delay and potential artifacts caused by the need to abruptly arrest the flow have made it difficult to achieve dead times shorter than about 1 ms. These complications can be avoided by continuously pumping the reactants through the mixer while monitoring reaction progress along a flow channel downstream from the mixing region [21,22,23]. More recent advances in mixer technology and detection methods [8,9,10,11,24,25,26,27,28,29,30] have resulted in major improvements in time resolution and sensitivity of continuous-flow instrumentation. By coupling continuous-flow mixing with a variety of detection methods, including tryptophan or tyrosine fluorescence, fluorescence resonance energy transfer (FRET), fluorescence life time, absorbance, circular dichroism, small-angle X-ray scattering and single-molecule spectroscopy, turbulent mixing devices have yielded a wealth of dynamic and structural information on early stages of protein folding on the microsecond-to-millisecond time scale [14,15,31,32,33,34,35,36,37,38,39,40,41,42,43,44,45,46,47,48,49,50,51,52].

## 2. Methods and Results

### 2.1. Design Criteria for Turbulent Mixers

At the molecular level, complete mixing of fluids is ultimately limited by diffusion. For example, the diffusion constant of urea used as a denaturant in many protein folding studies is 1.382 × 10^−9^ m^2^/s in water, indicating that the variance in molecular position after 10 μs of diffusion is 1.382 × 10^−2^ μm^2^ = (0.12 μm)^2^. Thus, mixing of low molecular weight solutes within 10 μs can be achieved only if the fluid components can be interspersed to within about 0.1 μm, and rapid mixing on this time scale requires submicron scale flow profiles. Turbulent flow mixers achieve nearly complete mixing of reactants by relying on the small fluid eddies generated under highly turbulent flow conditions [23,53]. The size of eddies is inversely related to the Reynold’s number and is thus a function of flow rate, viscosity and channel dimensions. Efficient mixing therefore requires high flow velocities, typically ~10 m/s, for micron scale mixers, and large sample volumes (1–10 mL).

Building on earlier design principles [23], Shastry et al. [8] developed a capillary mixer consisting of two concentric quartz tubes with a ~100-μm diameter platinum sphere placed near the exit of the inner capillary (Figure 1A, Mixer 1). To ensure precise positioning of the sphere, three quartz posts with a 10 μm diameter were fused to the inner surface of the outer capillary. The highly turbulent and uniform flow field in the wake of the sphere ensures rapid and complete mixing of the two solutions injected into the inner capillary and the space between the inner and outer capillaries, respectively. By forcing the solutions through a narrow (~10 μm) gap around the perimeter of the sphere, the size of eddies generated downstream is expected to be in the micron-to-submicron size range, resulting in diffusion-limited mixing times in the 10 to 100 μs range. Consistent with this estimate, the dead times we measured by using the quenching of tryptophan by N-bromosuccinimide (NBS) as a test reaction [17] were of the order of 50 μs [8,33], as illustrated in Figure 1A. Compared to earlier designs, which monitored reaction progress point by point on a free-flowing jet [9,23], we were able to greatly improve the reproducibility and sensitivity of continuous-flow measurements by mounting our capillary mixer on top of a quartz cuvette with a 0.25 × 0.25 mm^2^ flow channel and using a CCD-based fluorescence imaging setup to generate a continuous profile vs. distance downstream from the mixer. However, construction of these delicate mixers is challenging, and their performance characteristics, including dead time, backpressure and optical properties, can be quite variable.

### 2.2. Design of a Microfabricated Mixer

As a more robust approach to producing mixers, we relied on commercially available laser edging techniques (Translume, Inc., Ann Arbor, MI, USA) for microfabrication of channels with submicron tolerance in fused silica (quartz) [54]. Two related mixing chips based on a cross-channel design are illustrated in panels B and C of Figure 1. The mixers have three inlet ports and one outlet channel. Reactant P (e.g., unfolded protein) is injected into the central port, and reactant B (e.g., refolding buffer) is injected into each of the two side ports. The inlet ports are connected to the mixing chamber via a straight channel for the simple cross-mixer (Mixer 2) while a bottleneck-shaped channel for Mixer 3 is designed to increase the linear velocity of incoming flow to the chamber while keeping the backpressure low. The downstream observation channel of Mixer 3 has a conical shape in order to enhance the time resolution at short times after mixing and to extend the observable time window out to several milliseconds. At the same time, the increase in average channel width substantially lowers the backpressure for Mixer 3 (10–20 bar (1–2 Mpa) at flow velocities of 0.7 to 1.0 mL/s) compared to Mixer 2 (30 bar at 0.4 mL/s). In Figure 1 (lower panels), we compare the performance of these micro-fabricated mixers with that of our original capillary mixer by using the quenching of N-acetyltryptophan (NAT) fluorescence by NBS as a test reaction [17]. At a flow velocity of 1 mL/s, the observable time window of the capillary mixer ranges from the dead time (46 μs under these conditions) out to t_max_ ≈ 1 ms. The dead time of the simple cross-mixer (Mixer 2) was 8 ± 2.5 μs and t_max_ ≈ 1 ms at a flow velocity of 0.4 mL/s. Mixer 3 had a similar dead time (11.6 ± 0.9 μs) but an extended time window (t_max_ = 3 ms at 0.7 mL/s). A major advantage of the microfabricated mixers is reproducibility. Because of the submicron tolerance of the laser manufacturing process, the two lots of cross-mixers (Mixer 3) we compared showed virtually identical performance characteristics, confirming that the performance of microfabricated mixers is far more consistent compared to our previous hand-made capillary mixers.

### 2.3. Solution Delivery and Fluorescence Detection

To ensure constant, well-defined flow velocities for two reagents at variable mixing ratios, we replaced the pneumatic syringe drive used in our initial setup with a set of motor-driven syringe pumps (CETONI GmbH, Korbussen, Germany). The time axis of continuous-flow data can be accurately calculated from the known flow velocity and dimensions of the mixing channel, as detailed below.

To increase the sensitivity of fluorescence detection, we replaced the arc lamp and monochromator used in our previous setup [8] with a 30 W Q-switched diode-pumped solid-state laser (5 ns pulse duration, 5 μJ of single pulse energy) operating at 266 nm (Shanghai Dream Lasers Technology, Ltd., Shanghai, China). This intense and stable (<5% of power stability over 2 h) UV laser is well suited for excitation of tryptophan and tyrosine fluorescence in proteins at low concentrations. In earlier continuous-flow fluoresce experiments using powerful (350–500 W) Hg or Hg-Xe arc lamps for excitation, including our initial folding experiments on cytochrome c (cyt c) [8,32], we typically worked at protein concentrations of 20–40 μM (after mixing). Our current laser-based setup now yields data of comparable quality at protein concentrations as low as 1 μM, as documented below.

In previous laser-based continuous-flow instruments, the laser was focused on the flow channel and reaction progress was sampled point by point downstream from the mixing region [9,10,24,25]. Generating a complete kinetic trace using this approach requires dozens of separate measurements while maintaining continuous-flow conditions for extended periods of time, consuming large amounts of reagents. In contrast, we used a pair of Galvano mirrors and a cylindrical mirror to focus the laser onto the flow channel and control its position (Figure 2). The mirrors are regulated to scan the length of the flow channel using a triangular wave function, resulting in uniform distribution of light intensity vs. distance over a 23 mm segment of the channel. As in our original setup (Figure 1A), a quartz lens (f = 3.5 mm) and 305 nm high-pass interference filter (Semrock, Rochester, NY, USA) are used to project a magnified image of the flow channel onto a CCD detector (Figure 2). Thus, in a single continuous-flow experiment typically lasting 10 s (including a ~3 s acceleration time prior to activating the CCD camera), we can record a complete trace of fluorescence emission vs. time after mixing over the time range from t_d_ (~10 µs) to t_max_ (~2.5 ms) with an effective time resolution of ~1 µs in the narrow portion of the observation channel near the mixing region and ~4 µs near the end. We found empirically that a scanning frequencies of 20 Hz is optimal for achieving stable and uniform fluorescence excitation. This scan rate is negligibly slow compared to the pulse repetition rate of the laser but fast compared to the exposure times of ~10 s typically used in continuous-flow experiments. Thus, this novel excitation scheme is equivalent to continuous-wave illumination using a conventional light source but achieves more uniform and intense fluorescence excitation.

### 2.4. Data Collection and Analysis

The procedures for data collection and analysis using Mixer 3 are similar to those described by Shastry et al. [8], except for the calculation of the time axis. Because of the conical shape of the flow channel of Mixer 3, the reaction time at constant flow rate is no longer a linear function of the distance from the mixing region. At the flow rates typically used (0.5–1 mL/s), we estimate Reynolds numbers Re of 2300–4500, corresponding to moderately to strongly turbulent flow conditions. In this case, wall effects can be neglected, and the flow velocity profile, ***v****(**r**)*, can be approximately described as a 2-dimensional radial function of the position, ***r****(r, θ)*, within the widening flow channel of constant depth (Figure 3): (1)$$v\left(r\right)=\frac{V}{2\theta_{0}dr^{2}}r,$$
where *V*, *θ*_0_, *d*, *r* and *θ* are the volume velocity, opening angle of channel, channel depth and the magnitude and angle of the pointing vector ***r***, respectively (Figure 3A). Using Cartesian coordinate system with origin placed at the exit of the mixing region, the reaction time *t_r_* at each point is represented as follows: (2)$$t_{r}=\frac{\theta_{0}d}{V}({\left(x-x_{0}\right)}^{2}+y^{2})=\frac{\theta_{0}d}{V}(1+tan^{2}\left(\theta \right)){\left(x-x_{0}\right)}^{2},$$
where *x*_0_ is the position of the virtual vertex of the cone and $tan\left(\theta \right)=y/\left(x-x_{0}\right)$; thus, −*θ*_0_ < *θ* < *θ*_0_. Given the shallow open angle of the channel, $tan^{2}\left(\theta \right)$ is neglected (~7 × 10^−5^), and thus *t_r_* is effectively a function of *x* only. As a result, the position dependence of *t_r_* simplifies to:(3)$$t_{r}\left(x\right)\approx \frac{d}{V}\left(\frac{w_{L}-w_{0}}{2L}x^{2}+w_{0}x\right)+∆t,$$
where *w_L_* and *w*_0_ are the channel width at *x* = *L* and 0, respectively, and Δ*t* is the mixing time. The first term is consistent with the time required to fill the volume between [0, *x*] at constant flow rate *V*. 

To confirm the validity of Equation (3) for describing the nonlinear time axis of Mixer 3, we measured the quenching reaction of NAT fluorescence in the presence of excess NBS (Figure 3B). The kinetics of this pseudo first-order reaction is given by:(4)$$f\left(t,\left[NBS\right]\right)\approx f_{0}exp\left(-k^{\prime \prime }\left[NBS\right]t\right),$$
where *f*_0_ is the initial fluorescence intensity, and *k*″ is the second-order rate constants. Normally, the time course of reaction is analyzed by fitting to a single-exponential function at a fixed NBS concentration to obtain the apparent rate constant, *k*″[*NBS*]. In contrast, we evaluate Equation (4) at a fixed time (or a fixed pixel position) and variable NBS concentration (125 μM to 32 mM) in order to estimate *k*″*t* (Figure 3C,D). The *k*″*t* values obtained were a quadratic function of *x*, as predicted by Equation (3) (Figure 3E), thus confirming the validity of the underlying approximation. On the basis of the scatter of the points in panel E, we estimate that the error of the time-axis calibration is 0.3% at the earliest time points and 4% at long times.

### 2.5. Optimization of Mixing Conditions

Key considerations in optimizing mixing efficiency and performance are mixing ratio and flow rate. Higher mixing ratios are desirable for achieving large changes in solvent conditions but require more concentrated stock solutions, which may be limited by solubility. First, we used the NAT-NBS reaction to estimate the dead time of Mixer 3 at constant flow rate (0.7 mL/s) and variable mixing ratios (Figure 4A). The shortest dead time, 12 μs, was obtained when mixing 1 part of solution P (center port) with 10 parts of solution B (side ports; see Figure 1). Thus, Mixer 3 shows the best performance when reactions are initiated by a large concentration jump, which was one of our design criteria. Interestingly, the second-best dead time, 17 μs, was obtained for 1:1 mixing. With respect to flow rate optimization, higher flow velocities are expected to produce more turbulent flow in the mixing region, resulting in improved mixing efficiency and shorter dead times, although this comes at the cost of higher consumption of reagents and a shortened observation window. However, further increases in flow rate not only lead to a sharp increase in backpressure but may also give rise to light scattering artifacts due to cavitation, i.e., bubble formation due to the large pressure drop below the mixing region. By measuring the NAT-NBS reaction at different flow rates (1:10 mixing ratio), we found that the dead time of Mixer 3 decreases sharply with increasing flow rate, reaching values of about 20 μs at 0.7 mL/s to 1.0 mL/s. Under these conditions, we expect turbulent flow, based on estimated Reynolds numbers (3200 to 4500). At flow rates of 0.7 mL/s and above, the apparent rate of the NAT-NBS reaction increases linearly with NBS concentration up to 32 mM, as expected for a second-order reaction (Figure 4C). However, at flow rates less than 0.7 mL/s, the apparent rates at high NBS concentrations fall below those expected for a second-order reaction, indicating that the observed kinetics is increasingly limited by the rate of mixing under these sub-optimal flow conditions (Reynolds numbers less than 2400). In summary, optimal performance of Mixer 3, including dead times of 20 μs or less and a time window extending out to 2 to 3 ms, is achieved by using a mixing ratio of 1:10 and flow rates of 0.7 to 1.0 mL/s.

### 2.6. Mixing Efficiency

To quantify mixing efficiency, we relied on the quenching of NAT fluorescence by potassium iodide (Figure 5A). This collisional quenching process is fast compared to the time scale of mixing, and quenching efficiency is concentration-dependent (90% at 1 mM potassium iodide). Thus, the normalized decrease in NAT fluorescence upon mixing with iodide provides a direct measure of mixing efficiency. At a flow rate of 0.8 mL/s, the fluorescence intensity dropped to 14% of the original value at the first observable data point after the mixing region, indicating that the mixing efficacy is ~96%. At lower flow rates (0.6 and 0.4 mL/s), the residual fluorescence was much higher and decayed more slowly, resulting in substantially longer dead times (≥100 µs). At a flow rate of 1.0 mL/s, the initial fluorescence was slightly higher compared to 0.8 mL/s, most likely due to the onset of cavitation and concomitant light scattering artifacts. We also performed mixing efficiency tests in the presence of 8 M urea, which is commonly used in protein unfolding and refolding experiments (Figure 5B). The iodide quenching profiles obtained were similar to those in the absence of urea, indicating that high concentrations of urea do not impair mixing efficiency. This was confirmed by measuring the dead time, using the NAT-NBS reaction in the presence of 16.2% sucrose with viscosity matching that of an 8 M urea solution (note that NBS reacts with urea). To mimic conditions of a typical protein unfolding experiment, an aqueous solution of NAT was mixed with 10-fold excess of NBS dissolved in 17.9% sucrose, yielding a final concentration of 16.2%. As expected, the increase in viscosity results in a somewhat longer dead time (27 μs), but mixing efficiency remains high and mixing artifacts negligible even when mixing solutions of widely differing density and viscosity (Figure 6).

### 2.7. Accuracy

To assess the accuracy of the kinetics data obtained by Mixer 3, we measured the second-order rate, *k*″, of the NAT-NBS reaction by measuring the observed rate constant vs. NBS concentration under pseudo-first-order conditions (Figure 1C). The observed *k*″ of 1.2 × 10^6^ M^−1^ s^−1^ is in excellent agreement with that measured previously using a capillary mixer (Figure 1A). Taken together, these test results confirm that our microfluidic mixer design allows accurate and reliable measurements of rate constants as fast as ~10^5^ s^−1^. 

## 3. Applications

### 3.1. Folding of Cytochrome c

As a first application of our improved continuous-flow setup, we used Mixer 3 to monitor early stages of folding of cytochrome c (cyt c), which has long served as a test case for the development of new methodologies [4,5,31,32,55,56]. Horse cyt c is a 104-residue protein with a covalently attached heme group. The fluorescence of its sole tryptophan residue, Trp59, is governed by Förster energy transfer interaction with the heme, resulting in complete quenching in the folded state [57]. Prior studies using commercial stopped-flow instruments with a dead time of 1 ms or longer showed that the fluorescence change upon refolding accounts for at most 15% of the total fluorescence change, indicating that major conformational changes occur on the sub-millisecond time scale [58]. The development of a capillary mixing apparatus with a dead time of ~40 µs [8] made it possible to resolve the entire fluorescence change associated with the refolding of acid-denatured cyt c triggered by a jump in pH from 2.0 to 4.5, including a major phase with time constant τ_1_ = 41 μs accounting for 59% of the amplitude and a second phase with time constant τ_2_ = 648 μs accounting for 41% of the amplitude (green trace in Figure 7A; c.f. [32]). When we repeated this experiment using Mixer 3 (red trace), we also observed a biphasic decay in fluorescence with identical (within error) time constants (τ_1_ = 45 μs, τ_2_ = 605 μs) and similar amplitudes (A_1_ = 0.54 and A_2_ = 0.46). Minor differences in the amplitudes of the kinetic traces obtained with our new setup (Figure 1C) compared to the previous configuration (Figure 1A) can be explained by the fact that the ratio of tryptophan vs. tyrosine fluorescence excitation is lower at the wavelength of the UV laser (266 nm) compared to that used previously (280 nm). In both cases, a double-exponential fit accurately describes the observed fluorescence decay, and the extrapolated value at t = 0 is within ±5% the fluorescence of the acid-unfolded initial state, indicating that we are able to resolve the complete time course of folding associated with changes in intrinsic fluorescence. To document reproducibility, we compared the kinetic traces obtained in four independent experiments. The mean values and standard deviations for the corresponding kinetic parameters were as follows: <τ_1_> = 51 ± 14 μs; <A_1_> = 0.55 ± 0.09; <τ_2_> = 602 ± 51 μs; <A_2_> = 0.39 ± 0.05.

We have previously attributed the fast phase during refolding of cyt c in terms of a barrier-limited collapse of the polypeptide chain, based on its large amplitude consistent with a major decrease in average distance between the covalently attached heme and Trp59, as well as the temperature dependence of its rate constant indicative of a small but significant activation energy [32,33,59]. In a more recent study, using a microfluidic mixer to perform quenched H/D exchange measurements on the sub-millisecond time scale, we found that amide groups in two α-helices in the C-terminal half of cyt c were partially protected already during the fast (40 μs) phase, indicating that the initial compaction of the polypeptide chain involved formation of short-range helix–helix contacts, and is thus inconsistent with a random polymer collapse [60]. Helical structure in the N-terminal region begins to appear only during the second (600 μs) folding phase, suggesting that long-range tertiary contacts are established during the later stages of folding. Mitic et al. recently used an absorbance-detected continuous-flow mixing setup with a 4 µs dead time to monitor the kinetics of refolding of acid-denatured cyt c [11]. In addition to two kinetic phases with time constants similar to those of our two fluorescence-detected phases (83 and 345 μs, respectively), Mitic et al. observed an additional process with a time constant of 4.7 μs they attribute to binding of the His18 side chain to the initially four-coordinated heme iron. Because the heme group is covalently linked to the protein via thioester linkages to Cys 14 and Cys 17, this is a local conformational event not expected to be associated with a change in Trp59 fluorescence.

In addition to documenting the fact that the kinetic data obtained using Mixer 3 are fully consistent with those obtained previously using Mixer 1, Figure 7A also indicates a major improvement in the signal-to-noise ratio (S/N) and data quality. Our new excitation scheme using a 266 nm UV laser in combination with a set of Galvano mirrors (Figure 2) resulted in a high-quality kinetic trace with ~2-fold higher S/N on a 4 μM cyt c solution compared to that measured previously at a fivefold higher protein concentration using a Hg-Xe arc lamp and monochromator tuned to 280 nm. By repeating these experiments at several lower cyt c concentrations, we obtained reproducible kinetic traces with acceptable S/N down to protein concentrations as low as 0.5 μM (Figure 7B). Even though the 266 nm laser is suboptimal in terms of excitation of tryptophan fluorescence, comparison with the data obtained previously using 280 nm excitation (green trace in Figure 7A) indicates that our redesigned optical configuration has resulted in a ~20-fold increase in S/N.

### 3.2. Binding of a Peptide Ligand to a PDZ Domain

Rapid mixing methods have also played a central role in mechanistic studies of protein–ligand interactions. If ligand binding is coupled with a change in protein conformation, one can envision two limiting mechanisms, depending on whether the second-order binding step precedes the conformational change (“induced fit”) or occurs after formation of a binding-competent conformation (“conformational selection”). Both scenarios have been documented for different systems [61,62] or even for the same protein–ligand pair under different conditions [63,64]. One approach for distinguishing between these mechanisms is to measure the kinetics of binding as a function of either ligand or protein concentration [62]. The difference in predicted kinetics is especially pronounced at high ligand (or protein) concentrations where rate constants often exceed 1000 s^−1^ and thus require instrumentation for ultra-fast mixing [39]. As an illustrative example, we show in Figure 8 the kinetics of binding of a cognate peptide ligand to the first PDZ domain of NHERF1/EBP50 [65], a signaling adaptor at the membrane–cytoskeleton interface comprising a pair of PDZ domains and a C-terminal ezrin-binding motif. High-affinity binding of the ligand peptides studied here, CFTR_6_ (Ac-VQDTRL) and CFTR_10_ (Ac-TEEEVQDTRL), derived from the C-terminus of the cystic fibrosis transmembrane conductance regulator (CFTR), is primarily mediated by interactions of the C-terminal DTRL motif with a deep groove on the surface the PDZ domain [66].

To follow the time course of binding as a function of peptide concentration, we monitored the ligand-induced change in fluorescence of a tyrosine residue located in a pocket recognizing the C-terminal leucine of the CFTR peptide (Figure 8A). The rate constants observed under pseudo-first-order conditions increased sharply with increasing peptide concentration, reaching a value of 43,000 *±* 3000 s^−1^ (τ = 23 μs) at 3.5 mM (Figure 8B). The results document the ability of our instrument to resolve changes in inherently weak tyrosine fluorescence on the 10 μs time scale. Initially, the observed rate constant increases linearly with peptide concentration but begins to level off somewhat at concentrations above 1 mM. This behavior is intermediate between the strongly hyperbolic concentration dependence reported for PTP-BL PDZ2 and the strictly linear dependence on ligand concentration found for PSD-95 PDZ3 [39], indicating that these structurally closely related protein–ligand pairs experience different degrees of conformational changes and exhibit a surprising range of kinetic properties.

## 4. Conclusions

We have described the design and testing of an instrument for continuous-flow fluorescence measurements on the microsecond-to-millisecond time scale that combines a highly efficient microfluidic mixer/flow cell assembly on a quartz chip with a novel laser-scanning scheme for enhanced fluorescence excitation. The instrument features several performance characteristics that are critical for routine application to the ultrafast kinetic analysis of biomolecular folding and binding reactions. The dead time of 12 μs we measured under optimal flow conditions at low viscosity is at or near the low end of the fluorescence-based rapid mixing devices reported in the literature. Mixing efficiency remains high even under more viscous conditions, which is essential for studies of denaturant-induced protein folding or unfolding reactions. Further reductions in mixer and channel dimensions have been reported to result in somewhat shorter dead times, but this comes at the cost of major increases in backpressure and sample concentration. In contrast, by combining a tapered observation channel with a novel Galvano-mirror laser-scanning strategy, our instrument yields high-quality kinetic traces ranging from tens of microseconds to several milliseconds on protein concentrations as low as 1 µM. Representative kinetic data on early events during folding of cytochrome c and binding of a peptide ligand to a PDZ domain illustrate the power of the method for exploring the dynamics of macromolecular folding and binding reactions.

## Figures and Tables

**Figure 1 molecules-27-03392-f001:**
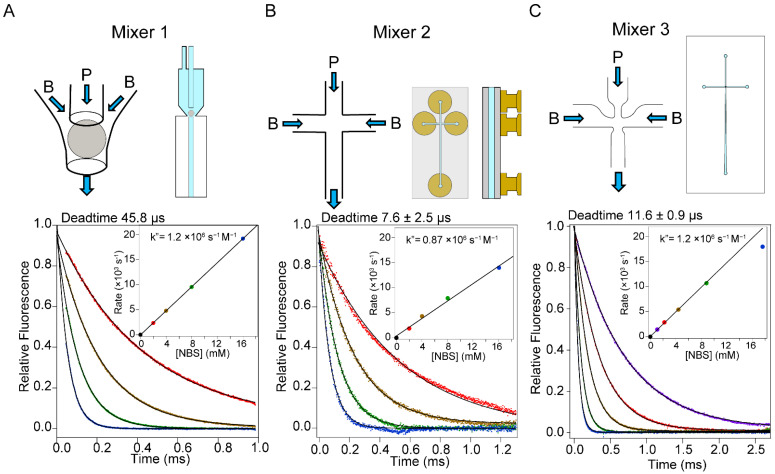
Summary of the mixer design and dead-time calibration of three continuous-flow mixing devices. Upper panels show schematics of the mixing regions and mixer/flow-cell assemblies for: (**A**) Mixer 1, the capillary mixer of Shastry et al. [8], (**B**) Mixer 2, a simple quartz cross-channel mixing chip (Translume, Inc., Ann Arbor, MI, USA) and (**C**) Mixer 3, a customized cross-channel mixing chip designed for reduced backpressure and expanded observation window (see Figure A1 in Appendix A for expanded diagram). The lower panels show representative examples of dead-time calibration using the pseudo-first-order NAT-NBS reaction. Typically, the fluorescent reagent (e.g., protein or NAT) is injected into central channel (*P*), and the second component (e.g., buffer, denaturant, ligand) is injected into the side channels *B*. The total flow rates in the experiments shown in A, B and C were 1 mL/s, 0.4 mL/s and 0.7 mL/s, respectively. The fluorescence decay curves are color-coded according to NBS concentration: purple for 1 mM, red for 2 mM, brown for 4 mM, green for 8 mM and blue for 16 m. Insets: determination of the second-order rate constant of the NAT-NBS reaction, based on the linear NBS concentration dependence of observed rate constants (determined by single-exponential non-linear least-squares fitting; solid lines).

**Figure 2 molecules-27-03392-f002:**
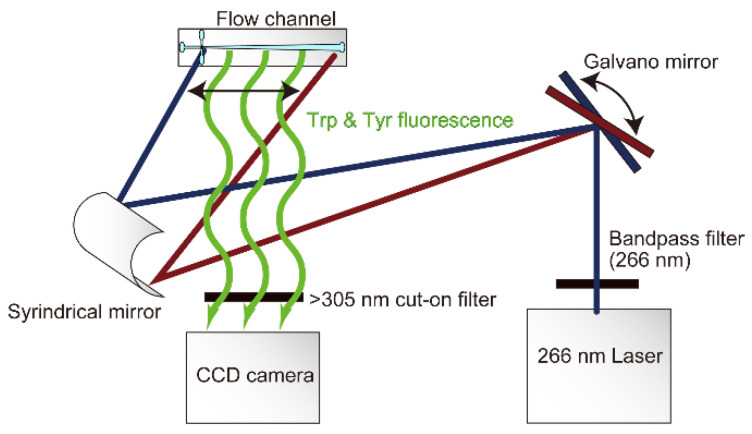
Optical configuration of our new continuous-flow instrument, using a UV laser in conjunction with a pair of Galvano mirrors to achieve uniform fluorescence excitation along the flow channel of the microfluidic mixing chip (Figure 1B,C).

**Figure 3 molecules-27-03392-f003:**
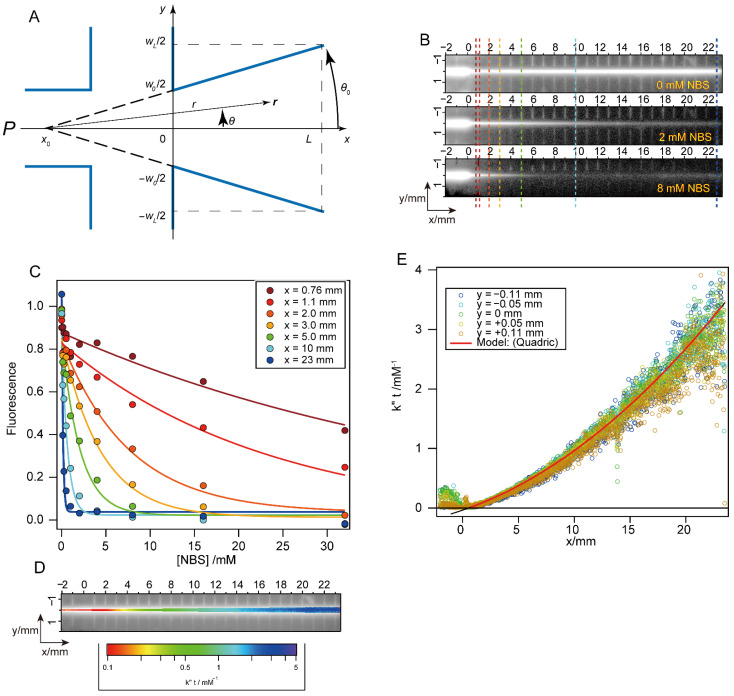
Calibration of the time axis of Mixer 3. (**A**) Schematic of the mixing region and conical flow channel (not to scale). The central inlet port (P in Figure 1) is on the left. The right exit is the observation channel. The x-axis is set along the observation channel with the origin on the entrance of it. *x*_0_ is the virtual vertex of the conical observation channel. The exit of the flow channel is located at *x* = *L*. The y-axis is set along the cross-section of the channel. The channel widths at the entrance (*x* = 0) and the exit (*x* = *L*) of the flow channel are *w*_0_ and *w_L_*, respectively. (**B**) Representative profiles of the NAT-NBS quenching reaction. The positions at *x* = 0.76 mm, 1.1 mm, 2.0 mm, 3.0 mm, 5.0 mm, 10 mm and 23 mm are shown by the lines in brown, red, orange-yellow, green-cyan and blue, respectively. (**C**) The concentration dependence of fluorescence intensity at representative positions. The data are fitted to Equation (4). (**D**) The profile of *k*″*t* value obtained by curve fitting in panel C. (**E**) Dependence of distance *x* on *k*″*t*.

**Figure 4 molecules-27-03392-f004:**
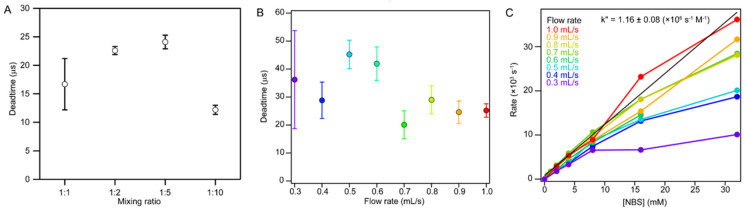
Optimization of mixing ratio and flow rate for Mixer 3. (**A**) Mixing ratio dependence and (**B**) flow rate dependence of dead time. (**C**) The NBS concentration dependence of rate constant obtained at several flow rates. The color code is shown in panel C.

**Figure 5 molecules-27-03392-f005:**
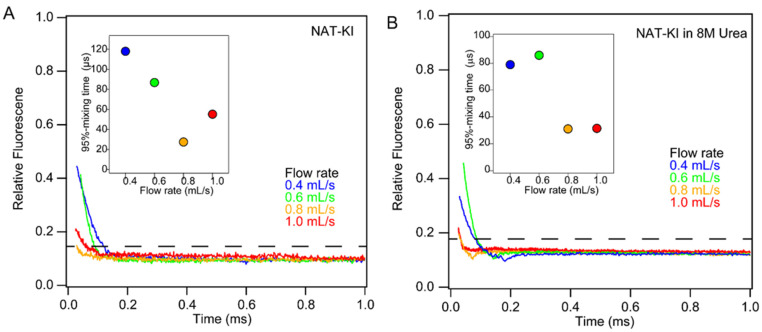
Mixing efficiency of Mixer 3 vs. flow rate assayed using potassium iodine quenching of tryptophan fluorescence. Residual fluorescence of NAT (10 mM) upon mixing with 1 mM potassium iodide in the absence (**A**) and presence (**B**) of 8 M urea. Color code: 0.4 mL/s (blue), 0.6 mL/s (green), 0.8 mL/s (orange) and 1.0 mL (red). The 95% mixing level is shown by the broken lines. The inlet panels show the time to achieve 95% mixing.

**Figure 6 molecules-27-03392-f006:**
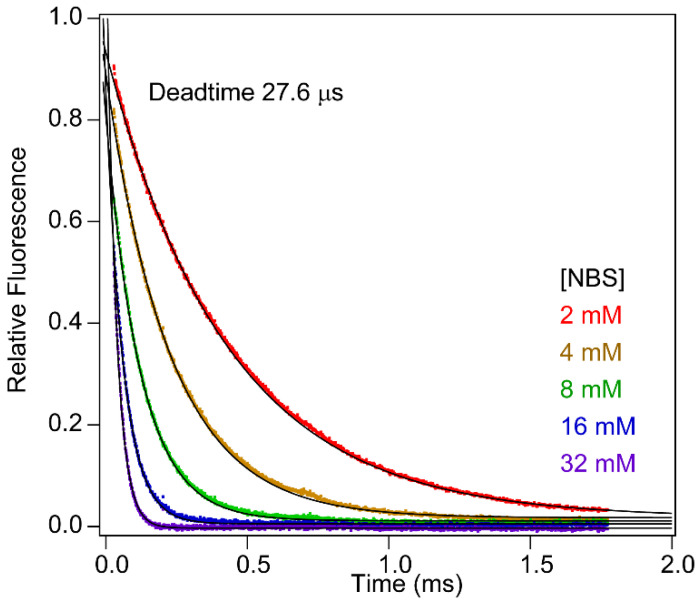
Dead-time calibration performed in the presence of sucrose. NBS is dissolved in 17.9% sucrose, which matches the viscosity of 8.8 M urea (η/η_0_ = 1.777). The viscosity of sucrose after 1:10 mixing (16.2%, η/η_0_ = 1.663) matches that of 8 M urea.

**Figure 7 molecules-27-03392-f007:**
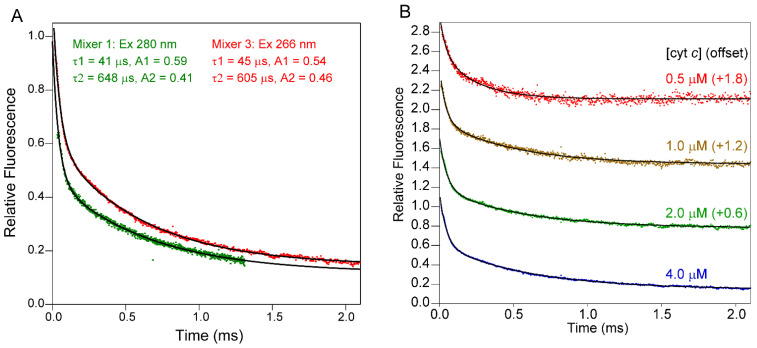
(**A**) Kinetics of refolding of acid-denatured cyt c (pH 2, salt-free) triggered by a jump to pH 4.5 (100 mM sodium acetate) recorded using Mixer 1 (green trace, 280 nm excitation) and Mixer 3 (red trace, 266 nm excitation). (**B**) Kinetics of refolding of cyt c (same conditions as in panel (**A**)) at different protein concentrations, as indicated.

**Figure 8 molecules-27-03392-f008:**
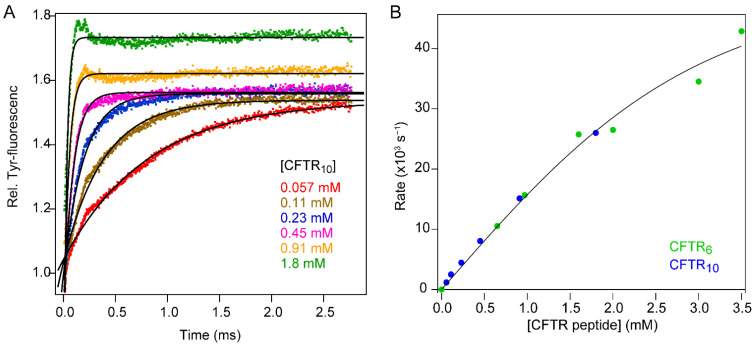
Binding kinetics of peptides from the C-terminus of CFTR to the first PDZ domain of NHERF1. (**A**) Kinetic traces of CFTR_10_ binding to PDZ1 of NHRF1. (**B**) Rate constants of the binding reaction for CFTR_10_ (blue) and CFTR_6_ (green). The line represents a hyperbolic fit of the rate constant vs. ligand concentration.

## Data Availability

Not applicable.

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
