# Peer review of "Advances in Mixer Design and Detection Methods for Kinetics Studies of Macromolecular Folding and Binding on the Microsecond Time Scale"

_molecules, 2022, doi:10.3390/molecules27113392_

Round 1

Reviewer 1 Report

In the manuscript titled “Advances in mixer design and detection methods for kinetics studies of macromolecular folding and binding on the microsecond time scale”. The authors are interested in exploring the mixer design and detection methods in microsecond time scale to understand protein folding and ligand binding. The paper is well written but I do recommend for few clarifications before publication.

Major

1) The methodology is used for only one test case? How it is approachable to the entire protein database. The authors should have done couple of different known and stable proteins and ligand systems to do these experiments to increase the strength of the manuscript.

2) How the method is significant/advantageous from methods that is being used for minimal sampling and minimal time experiments like MST(https://www.sciencedirect.com/science/article/pii/S0022286014002750#f0040). This is one example; I do recommend the authors to compare and contrast to reported experiments. This will definitely increase the value of the work.

3) A separate section describing the challenges under the topic of interest and how it is being overcome by different group in the past and the authors expertize on this field about their current protocols will help the readers to gain more out of this manuscript.

4) How is the timescale error in this experiment?

Minor

  • Figure resolution needs improvement.
  • It would be great if a separate flow chart or more details under data collection and analysis
  • How this will be useful to study non-structured proteins (NSP)
  • Will this answer about peptide assembly eg., antimicrobial peptides.
  • Conclusion needs to be more elaborate.

Author Response

In the manuscript titled “Advances in mixer design and detection methods for kinetics studies of macromolecular folding and binding on the microsecond time scale”. The authors are interested in exploring the mixer design and detection methods in microsecond time scale to understand protein folding and ligand binding. The paper is well written but I do recommend for few clarifications before publication.

Response: We appreciate the reviewer’s careful reading of the paper and suggestions for improving the paper.

Major

1) The methodology is used for only one test case? How it is approachable to the entire protein database. The authors should have done couple of different known and stable proteins and ligand systems to do these experiments to increase the strength of the manuscript.

Response: To illustrate and validate the enhanced rapid mixing technique presented in this paper, we chose to present two representative examples:
(a) The pH-induced folding reaction of acid-denatured cytochrome c, which is characterized by two kinetic phases on the microsecond-to-millisecond time scale, including major decay in tryptophan fluorescence with a time constant of 50 us that cannot be resolved with conventional techniques, such as stopped-flow. To test the sensitivity limits of the technique, we studied this reaction as a function of protein concentration, and showed that quality kinetic data can be obtained at concentrations as low as 1 uM (lower than any published results we are aware of).
(b) To illustrate applications to studies of protein-ligand binding mechanisms, we measured the kinetics of binding of a peptide ligand to a PDZ domain as a function of peptide concentration, up to the mM range where reaction rates exceed 40,000 s-1. As this protein lacks tryptophan residues, this study also demonstrates, for the first time, our ability to use the less sensitive tyrosine as a fluorescence probe.

We feel that these examples provide challenging test cases and are sufficient for  illustrating applications to two diverse areas of research. Sub-millisecond kinetic data have been published by us and others on folding reactions of numerous proteins, as well as a few binding reactions. Each of these studies is a major undertaking, and applications of our improved rapid mixing instrument to additional proteins are beyond the scope of this methods paper.

2) How the method is significant/advantageous from methods that is being used for minimal sampling and minimal time experiments like MST(https://www.sciencedirect.com/science/article/pii/S0022286014002750#f0040). This is one example; I do recommend the authors to compare and contrast to reported experiments. This will definitely increase the value of the work.

Response: While microscale thermophoresis (MST) is indeed a powerful technique for studies of macromolecular binding and other conformational changes, it is inherently slow, being limited by diffusion across a thermal gradient, and has primarily been used for equilibrium binding and folding studies. In contrast, the continuous-flow mixing technique described here is designed for kinetic studies of very fast reactions occurring over the microsecond to millisecond time range.

3) A separate section describing the challenges under the topic of interest and how it is being overcome by different group in the past and the authors expertize on this field about their current protocols will help the readers to gain more out of this manuscript.

Response: Throughout our manuscript, we compare and contrast our technique with alternative methods in the literature. In the introduction we list and provide references for the leading techniques for ultrafast kinetic studies, and discuss their strengths and limitations. We also mention challenges and provide a detailed rationale for the development of more efficient mixers and more sensitive detection methods. In the section on the design and testing of our device, we compare the results directly with those reported previously (e.g., Fig. 1). In Chapter 3 (Applications), we again present a detailed comparison of our new data with published results. Finally, under Conclusions we provide a concise summary of the key features, advantages and limitations or our technique. Thus, we feel that all the topics mentioned by the reviewer have been covered adequately.

4) How is the timescale error in this experiment?

Response: Thank you for this astute observation. At the end of Chapter 2.4. (p. 7) we added the following sentence:

On the basis of the scatter of the points in panel E, we estimate that the error of the time-axis calibration is 0.3% at the earliest time points and 4% at long times.

Minor

  • Figure resolution needs improvement.

We uploaded the figures at a resolution of 600 dpi, which should be sufficient. Editor: please let us know if we need to increase resolution (e.g., by providing the original Adobe Illustrator files).

  • It would be great if a separate flow chart or more details under data collection and analysis.

The rather involved procedures for data collection and analysis are described in detail in the text and illustrated in Figure 3. We tried, but were unable to come up with a meaningful flow chart.

  • .How this will be useful to study non-structured proteins (NSP)

While kinetic folding studies are naturally limited to globular proteins, the technique can be applied to study binding for systems where on or both binding partners are intrinsically unfolded.

  • Will this answer about peptide assembly eg., antimicrobial peptides.

This is a possible area of application, although the presence of lipid membranes in the case of antimicrobial peptides would pose a challenge, and these processes are likely to occur on a slower time scale covered by stopped-flow or manual mixing techniques.

  • Conclusion needs to be more elaborate.

As suggested, we have expanded the Conclusions section (p. 12).

Reviewer 2 Report

It was a pleasure to read this manuscript on the advances on mixing designs to study reaction kinetics. Heiner Roder did already a major contribution on this subject in the late 90s and I found the new developments really nice and useful. 

As a reader (like I did in the past...), I will borrow the ideas of the authors and try to copy their devices. I congratulate with them.

Manuscript worth publication for sure

Author Response

We appreciate the reviewer's strong endorsement and kind words!

Reviewer 3 Report

The authors describe designs of capillary mixing devices with mixing times in the microsecond range enabling to measure rate constants as large as 105 s-1. The instruments were tested by measuring conformational folding of cytochrome C induced by the pH jump from pH 2.0 to pH4.5. Two characteristic folded states have been identified, which were in line with known conformational dynamics of the protein. The devices were also tested for probing of the binding kinetics measured by the interaction of PZT protein with a cognate peptide binding. Overall, the description of the instruments is clear, so the paper can be published. I have a few minor comments.

  1. Conformational transition of Cyt C. Although the results obtained on two mixers reveal two different characteristic times, both data set are different; are these statistically different values? The results for only single experiments are shown. What is the reproducibility of the data obtained for both instruments? What conformational transitions of the protein do these times correspond to?  
  2. Similar the data accuracy and reproducibility concerns are addressed to the experiments with PZT protein and these should be clarified. What is the peptide in these experiments? A sequences needs to be indicated as well.

Author Response

  1. Conformational transition of Cyt C. Although the results obtained on two mixers reveal two different characteristic times, both data set are different; are these statistically different values? The results for only single experiments are shown. What is the reproducibility of the data obtained for both instruments? What conformational transitions of the protein do these times correspond to?  

Response: Thank you for these excellent suggestions. It turns out that, after correcting an error in the time axis calibration, the kinetic traces obtained with Mixers 1 and 3 we compare in Figure 7A are very similar. The time constants of the two kinetic phases are well within experimental error, and minor differences in relative amplitudes are explained by the difference in excitation wavelengths used for these two experiments, as explained at the bottom of p. 9. To document reproducibility, we performed four repeats of the folding experiments on Mixer 3 and listed the means and standard deviations of the corresponding kinetic parameters on p. 10 (top paragraph). In the second paragraph on p. 10, we now provide additional details on the nature and structural properties of the conformational transitions we observed within the first few milliseconds of folding of cytochrome c.

  1. Similar the data accuracy and reproducibility concerns are addressed to the experiments with PZT protein and these should be clarified. What is the peptide in these experiments? A sequences needs to be indicated as well.

Response: Because of limited amounts of reagents available, we were unable to perform multiple repeats of the binding experiments in Figure 8. However, we do not expect these traces to be any less reproducible than the cytochrome c folding experiments. This is borne out by the smooth concentration dependence of the rate constants in Fig. 8B, as well as the small fitting errors of the individual traces (e.g., see p. 12, top).

We now list the full sequence of the two CFTR peptides studied on p. 11 (above Fig. 8).

Round 2

Reviewer 1 Report

I do recommend for publication.